# Generation of Pseudo-Random Quantum States on Actual Quantum Processors

**DOI:** 10.3390/e25040607

**Published:** 2023-04-03

**Authors:** Gabriele Cenedese, Maria Bondani, Dario Rosa, Giuliano Benenti

**Affiliations:** 1Center for Nonlinear and Complex Systems, Dipartimento di Scienza e Alta Tecnologia, Università degli Studi dell’Insubria, Via Valleggio 11, 22100 Como, Italy; 2Istituto Nazionale di Fisica Nucleare, Sezione di Milano, Via Celoria 16, 20133 Milano, Italy; 3Istituto di Fotonica e Nanotecnologie, Consiglio Nazionale delle Ricerche, Via Valleggio 11, 22100 Como, Italy; 4Center for Theoretical Physics of Complex Systems, Institute for Basic Science (IBS), Daejeon 34126, Republic of Korea; 5Basic Science Program, Korea University of Science and Technology (UST), Daejeon 34113, Republic of Korea; 6NEST-CNR Istituto Nanoscienze, 56126 Pisa, Italy

**Keywords:** quantum computing, NISQ devices, random quantum circuits

## Abstract

The generation of a large amount of entanglement is a necessary condition for a quantum computer to achieve quantum advantage. In this paper, we propose a method to efficiently generate pseudo-random quantum states, for which the degree of multipartite entanglement is nearly maximal. We argue that the method is optimal, and use it to benchmark actual superconducting (IBM’s *ibm_lagos*) and ion trap (IonQ’s *Harmony*) quantum processors. Despite the fact that *ibm_lagos* has lower single-qubit and two-qubit error rates, the overall performance of *Harmony* is better thanks to its low error rate in state preparation and measurement and to the all-to-all connectivity of qubits. Our result highlights the relevance of the qubits network architecture to generate highly entangled states.

## 1. Introduction

Quantum computers working with approximately 50–100 qubits could perform certain tasks beyond the capabilities of current classical supercomputers [1,2], and quantum advantage for particular problems has recently been claimed [3,4,5], although later simulations on classical supercomputers [6,7] have almost closed the quantum advantage gap. As a general remark, quantum advantage can only be achieved if the precision of the quantum gates is sufficiently high and the executed quantum algorithm generates a sufficiently large amount of entanglement that can overcome classical tensor network methods [8]. Therefore, for quantum algorithms, multipartite (many-qubit) entanglement is the key resource to achieve exponential acceleration over classical computation. Unfortunately, existing noisy intermediate-scale quantum (NISQ) devices suffer from various noise sources such as noisy gates, coherent errors, and interactions with an uncontrolled environment. Noise limits the size of quantum circuits that can be reliably executed, so achieving quantum advantage in complex and practically relevant problems is still a formidable challenge. It is therefore important to benchmark the progress of currently available quantum computers [9,10,11].

Quantifying entanglement is a demanding task [12,13]. In particular, the characterization of multipartite entanglement is not a simple matter, since, as the number of subsystems increases, we should consider all possible non-local correlations among parties in order to obtain a complete description of entanglement. Moreover, tomographic state reconstruction requires a number of measures that grows exponentially with the number of qubits [14]. Finally, there is no unique way to characterize multipartite entanglement [13].

On the other hand, bipartite entanglement can be probed by means of entanglement entropies. In particular, we can consider the *second order Rényi entropy* of the reduced density matrix for any of the subsystems. If it is larger than the entropy of the entire system, we can conclude that bipartite entanglement exists between the two subsystems. If the overall state is pure, the second-order Rényi entropy is directly a measure of bipartite entanglement. In that case, in order to quantify the amount of multipartite entanglement, one can look at the distribution of the Rényi entropy of a subsystem over all possible bipartitions of the total system. For example, Facchi et al. proposed [15] a method based on the probability density of bipartite entanglement between two parts of the total system; one expects that multipartite entanglement will be large when bipartite entanglement is large and does not depend on the bipartition, namely when its probability density is a narrow function centered at a large value.

Computing entanglement entropies requires the knowledge of the density matrix of the system. Unfortunately, probing the density matrix is also a challenging problem, especially as the dimension of the system increases. For this reason, it is necessary to indirectly estimate the entropy, for instance, using the method proposed by Brydges et al. [16] via randomized measurements.

For random pure quantum states, the entanglement content is almost maximal and the purity (and so the second-order Rényi entropy) probability distribution is well known. Unlike simpler states such as W and GHZ, for which the entanglement content is essentially independent of the dimension of the system, for random states, the average multipartite entanglement is an extensive quantity. Moreover, random states are relevant in the study of the complexity of quantum circuits [17] and black holes [18] and for benchmarking quantum hardware [9,19].

The purpose of this paper was to investigate strategies to efficiently generate highly entangled states and then find a way to quantify the actual amount of entanglement achieved in state-of-the-art quantum hardware. In particular, we propose a method (hereafter referred to as *direct method*) to efficiently generate pseudo-random quantum states for *n* qubits, approximating true random states to the desired accuracy by means of layers where a random permutation of the qubits is followed by two-qubit random state generation. We provide numerical evidence that this method converges to true *n*-qubit random states by increasing the number of layers as fast as the circuit implementing two-qubit random unitary gates using the KAK parametrization of SU(4) (*KAK method*) [20], but with reduced cost in terms of the number of CNOT gates. We also argue that the proposed method is optimal for pseudo-random quantum state generation. Finally, we implement the method to benchmark actual quantum processors. In particular, two different realizations of quantum hardware are compared: IBM’s superconducting-based devices and IonQ’s trapped ion-based devices. We show that, despite the fact that superconducting devices have smaller error rates than IonQ for one- and two-qubit gates, the overall performance is better in trapped ion devices. This is mainly due to the complete connectivity of these machines, which allows avoiding noisy SWAP gates to implement qubit permutations. Our results highlight the importance of quantum hardware architecture in the implementation of quantum algorithms.

This paper is organized as follows. In Section 2, we discuss and compare methods for the generation of pseudo-random states. In Section 3, we apply the direct method in real quantum hardware and compare the results for IBMQ and IonQ devices with the second-order Rényi entropy estimated via the method of Ref. [16]. Finally, our conclusions are drawn in Section 4.

## 2. Generation of Pseudo-Random Quantum States

In this section, we briefly discuss methods of generating pseudo-random states, starting with the exact strategy and ending with our proposal, which will be numerically verified by comparison with the standard KAK method.

Let |ψ〉 be a pure state that belongs to the Hilbert space $H=H_{A}⊗H_{B}$, where $H_{A}$ and $H_{B}$ are spanned, respectively, by ${\left\{|i_{A}\rangle \right\}}_{1\leq i_{A}\leq N_{A}}$ and ${\left\{|i_{B}\rangle \right\}}_{1\leq i_{B}\leq N_{B}}$. *A* and *B* are two bipartitions of the entire system. Assuming that, without loss of generality, $N_{A}\leq N_{B}$, the state admits a Schmidt decomposition [1]: (1)$$|\psi \rangle =\sum_{i=1}^{N_{A}}\sqrt{x_{i}}|a_{i}\rangle ⊗|b_{i}\rangle ,$$
where $\left\{|a_{i}\rangle \right\}$ and $\left\{|b_{i}\rangle \right\}$ are suitable orthonormal sets for $H_{A}$ and $H_{B}$, respectively, which depend on the particular state |ψ〉, and the scalars $x_{i}$, known as the Schmidt coefficients for |ψ〉, are real, non-negative, and unique up to reordering. These coefficients can be used to quantify the bipartite entanglement via the second-order Rényi entropy
(2)$$S^{\left(2\right)}\left(\rho_{A}\right)=-\log_{2}\left[R\left(\psi \right)\right],$$
with the reduced purity R(ψ) of the state given by
(3)$$R\left(\psi \right)=\mathrm{Tr}\left(\rho_{A}^{2}\right)=\sum_{i=1}^{N_{A}}x_{i}^{2},$$
where $\rho_{A}$ is the reduced density matrix (with respect to $H_{B}$) of the overall state ρ: (4)$$\rho_{A}={\mathrm{Tr}}_{B}\left(\rho \right)={\mathrm{Tr}}_{B}\left(|\psi \rangle \langle \psi |\right).$$

Hereafter, we shall focus on the purity, which is trivially related to the second-order Rényi entropy.

In the case of a random state, the cumulants of the purities’ probability distributions can be exactly calculated [21,22], more details on which can be found in Appendix A. In particular, the first cumulants are given by
(5)$$\mu_{N_{A}N_{B}}\equiv \langle R\rangle =\frac{N_{A}+N_{B}}{1+N_{A}N_{B}},$$
(6)$$\sigma_{N_{A}N_{B}}^{2}\equiv {\langle \left(R-\langle R\rangle \right)\rangle }^{2}=\frac{2\left(N_{A}^{2}-1\right)\left(N_{B}^{2}-1\right)}{{\left(1+N_{A}N_{B}\right)}^{2}\left(2+N_{A}N_{B}\right)(3+N_{A}N_{B})\prime }$$
and they will be used later to verify the quality of random state generation.

In order to generate a true *n*-qubit random state, the ideal (and only) rigorous way would be to apply a random unitary operator, with respect to the Haar measure of the unitary group $SU(N=2^{n})$ (neglecting the global phase of no physical significance). Unfortunately, the implementation of such an operator acting on the *n*-qubit Hilbert space requires a number of elementary quantum gates that is exponential in the number of qubits [1].

On the other hand, it has been proven that the sequences of random single qubit gates followed by a two-qubit local interaction (which can be an SU(4) random unitary operator, or more simply a single CNOT gate) generate pseudo-random unitary operators which approximate, to the desired accuracy, the entanglement properties of true *n*-qubit random states [23,24,25,26,27]. However, the random SU(4) strategy depicted in Figure 1, used for example in [9], performs better than a single CNOT in terms of convergence rate [28], with the cost of using three CNOTs instead of just one, as we will see below.

Hence, the problem now turns to find an efficient way (in sense that will be clarified later) to generate random SU(4) operators.

To this end, one possible strategy would be to use Hurwitz’s parametrization of the unitary group, SU(N), for the specific case of N=4 [25,29]. However, this approach has the disadvantage of requiring a large number of CNOTs—16 for the particular case of SU(4)—which are usually the main source of errors in NISQ devices [30].

### 2.1. Cartan’s KAK Decomposition of the Unitary Group

An alternative approach consists of using the Cartan’s KAK decomposition of a semi-simple Lie group G (in this case $SU(2^{n})$) which parametrizes the group in terms of the subgroups’ elements [31]. The case of SU(4) of interest here is described in Appendix B, and is the optimal construction [20] to implement a generic two-qubit gate, using at most 3 CNOT and 15 single-qubit gates.

### 2.2. Direct Generation of Two-Qubit Random Quantum States

Is the Cartan decomposition the most efficient way to generate a two-qubit random state? Let us think in terms of free parameters. The Cartan’s KAK decomposition is the optimal (in terms of the number of CNOT and single-qubit rotations) way to construct random SU(4) operators via quantum circuits. It requires 15 single-qubit rotations and so 15 independent real parameters (as one expects, since dim(SU(4))=15). On the other hand, a normalized random two-qubit state |ψ〉 depends, up to a global phase, on six independent real parameters. This suggests that, in some ways, it could be possible to build any two-qubit state (starting from some fiducial state) with at most six independent rotations and one CNOT (needed to entangle the system).

This expectation is confirmed [21] by the quantum circuit depicted in Figure 2, producing |ψ〉 from an initial state |00〉. How can this circuit be achieved? Starting from a state |ψ〉, and transforming it by the inverse of the circuit of Figure 2, one can end up with |00〉, specifying how the angles θ are obtained. Any two-qubit state can in fact be written, using the Schmidt decomposition as a sum of two product terms: (7)$$|\psi \rangle =\sqrt{x_{1}}|a\rangle |b\rangle +\sqrt{x_{2}}{|a\rangle }^{⊥}{|b\rangle }^{⊥},$$
where |a〉 and |b〉 are single-qubit states (of the first and second qubit, respectively), and ${|a\rangle }^{⊥}$ and ${|b\rangle }^{⊥}$ are single-qubit states orthogonal to |a〉 and |b〉, respectively (i.e., $\left\{|a\rangle ,{|a\rangle }^{⊥}\right\}$ and $\left\{|b\rangle ,{|b\rangle }^{⊥}\right\}$ are the Schmidt bases of the Hilbert spaces of the two qubits). The idea is, starting from this decomposition, to obtain the state |00〉 using unitary operations, and then, taking the inverse transformation, one can obtain the desired result. The angle $\theta_{4}$ is chosen such that the $R_{z}$ rotation of angle $-\theta_{4}$ eliminates a relative phase between the coefficients of the expansion of |a〉 into |0〉 and |1〉 (note that, because the circuit is considered in the reverse direction, the angles of rotations have opposite signs). A subsequent $R_{y}$ rotation with angle $-\theta_{3}$ results in the transformation |a〉→|0〉 (up to a global phase). Similarly, rotations of angles $-\theta_{6}$ and $-\theta_{5}$ rotate |b〉 into |0〉. After applying rotations of angles $-\theta_{3}$, $-\theta_{4}$, $-\theta_{5}$, and $-\theta_{6}$, the state has become, up to a global phase, of the form: (8)$$|\psi \rangle =\cos\theta_{1}|00\rangle +e^{i\theta_{2}}\sin\theta_{1}|11\rangle .$$

Finally, the $R_{z}$ rotation of $-\theta_{2}$ eliminates the relative phase between |00〉 and |11〉. The CNOT brings the second qubit to |0〉, and the last rotation of angle $\theta_{1}$ on the first qubit yields the final state |00〉.

In order to obtain a random state, it is necessary to know how to randomly sample the various angles $\theta_{i}$, that is, it is necessary to know their probability distributions with respect to some measure of the state space, associated with the parametrization provided by Figure 2. Formally, a quantum state |ψ〉 can be considered as an element of the complex projective space $CP^{N-1}$, with $N=2^{n}$ being the Hilbert space dimension for *n* qubits [32]. The natural Riemannian metric on $CP^{N-1}$ is the Fubini–Study metric induced by the unitarily invariant Haar measure on U(N). This is the only metric invariant under unitary transformations. Thus, the unnormalized joint probability distribution is simply obtained by calculating the determinant of the metric tensor with the parametrization [33]. The idea is to use these more efficient “operators” *D* to construct the *n*-qubit pseudo-random states, although formally they do not map the entire Bloch sphere if the initial state is not |00〉. Consider, for example, a dimensionally simpler case: from the north pole of a sphere, it is possible to reach any other point by making only two rotations. Thus, carefully choosing the distribution of the rotation angles, it is possible to uniformly map every point of the sphere, but this is no longer valid if the starting point is changed, where the worst case scenario is a point on the equator.

In general, one expects that the error committed in sampling the Bloch space is small and everything converges to a random state anyway (see below). In Figure 3, the circuit used to generate the random state (in a similar way to Figure 1) with this method is shown, which will be referred to from now on as the direct method.

### 2.3. Comparison of KAK and Direct Method

In general, for an *n*-qubit state, there are nna ways to construct a bipartition in $n_{a}$ and $n_{b}=n-n_{a}$ qubits ($N_{A}=2^{n_{a}},N_{B}=2^{n_{b}}$). Clearly, $n_{a}$ can be any natural number from 1 to *n*. For the sake of clarity, let us consider, for example, a four-qubit state, wherein the qubits are labeled {0,1,2,3}. The $n_{a}=2$ bipartition can be obtained in 42=6 different ways by tracing out the pair of qubits {0,1}, {0,2}, {0,3}, {1,2}, {1,3} or {2,3}. In the case of a random state, these partitions are equivalent, i.e., the value of purity is independent of the choice of the subset of qubits traced out.

Given a quantum state generated as shown in Figure 1 (KAK method) and Figure 3 (direct method), and taking an ensemble of $N_{e}$ states, we numerically estimate the mean value ${\left(\mu_{2^{n_{a}}2^{n-n_{a}}}\right)}_{e}$ and the variance ${\left(\sigma_{2^{n_{a}}2^{n-n_{a}}}^{2}\right)}_{e}$ of the purities of the generated pseudo-random quantum state. Simulations are performed using the Python library *Qiskit*, particularly the system density matrix, which is computed using the built-in state–vector simulator. In order to evaluate how well the states are generated, the idea is to calculate the relative error of the mean value $\Delta_{\mu }$ and the variance $\Delta_{\sigma^{2}}$, which are averaged over each possible bipartition of the number of qubits:(9)$$\bar{\Delta_{\mu }}=\frac{1}{n-1}\sum_{n_{a}=1}^{n-1}\frac{|{\left(\mu_{2^{n_{a}}2^{n-n_{a}}}\right)}_{e}-\mu_{2^{n_{a}}2^{n-n_{a}}}|}{\mu_{2^{n_{a}}2^{n-n_{a}}}},$$
(10)$$\bar{\Delta_{\sigma^{2}}}=\frac{1}{n-1}\sum_{n_{a}=1}^{n-1}\frac{|{\left(\sigma_{2^{n_{a}}2^{n-n_{a}}}^{2}\right)}_{e}-\sigma_{2^{n_{a}}2^{n-n_{a}}}^{2}|}{\sigma_{2^{n_{a}}2^{n-n_{a}}}^{2}}.$$

The sum on $n_{a}$ is up to n−1, since $n_{a}=n$ means tracing out the whole system, i.e., calculating the purity of the whole state, which, being pure, has a unit mean and zero variance. In the previous formulas, the quantities $\mu_{2^{n_{a}}2^{n-n_{a}}}$ and $\sigma_{2^{n_{a}}2^{n-n_{a}}}^{2}$ are the expected values for a true random state, as shown in Equations (Equation 5) and (Equation 6), respectively.

The averaged relative error for the mean and variance is plotted as a function of the number of steps (i.e., of layers) of the generating circuit and the size of the statistical ensemble, for the cases of n=4, 6, and 8 qubits. Numerical data from Figure 4 suggest that the direct and the KAK methods are basically equivalent in terms of speed of convergence to the expectation values of random states. Notice that the number of steps required for convergence grows as ∼n, since at least n(n−1)/2 two-qubit gates are required in order to entangle all qubit pairs, and for each step, n/2 two-qubit gates are applied.

In Figure 5, the mean value and the variance of the purities are shown as a function of the number of qubits in a partition $n_{a}$ for systems with different size *n*. The moments are estimated considering an ensemble of $N_{e}=100$ pseudo-random states generated using the direct method with 20 steps.

The convergence to the true random state expected values improve as the dimension of the system increases. Indeed, for higher dimensions, the entanglement content is highly typical, i.e., it is possible to show that the entanglement distribution for a random state becomes strongly peaked in the limit of a large number of qubits. This concentration of the measure explains the better convergence for higher-dimensional cases [34].

## 3. Results on Actual Quantum Hardware

The circuits we implemented on real quantum hardware (IBM’s *ibm_lagos* and IonQ’s *Harmony*, a visualization of which is given in Figure 6) are slightly different from that shown in Figure 3. First of all, given the available resources, only circuits with four and six qubits were considered. In order to limit circuit depth, the random permutation gates are avoided, and instead, since all the qubits must be entangled with each other, the *D* (or SU(4)) gates are applied to qubit pairs labeled as {(0,1), (2,3), (0,2), (1,3)} (for the four-qubit case) and {(0,1), (2,3), (4,5), (1,2), (3,4), (0,5), (0,3), (1,4), (5,2)} (for the six-qubit case). The purities of a random state are estimated using measurements along randomly rotated axes, following the method proposed by Brydges et al. [16].

The ensemble of random states is $N_{e}=10$ wide, and for each state, $N_{m}=20$ random measurement axes are taken in order to estimate the purities. Each of these 200 circuits is followed by a measurement in the standard computational basis, and each circuit is repeated $N_{s}=1000$ times (number of shots, limited by the available budget for IonQ) in order to estimate the outcome probabilities of each element of the computational basis for each circuit.

Note that IBM’s quantum computers are nominally calibrated once over a 24-h period, and the system properties update once this calibration sequence is complete. Calibration plays a critical role in quantum circuit execution, since the properties of the systems are utilized for noise-aware circuit mapping and optimization (transpilation). Due to the daily calibration, it is difficult to compare the results obtained in different days on the same hardware. For this reason, all comparative results with the same quantum computer herein were taken with the same calibration data (i.e., the same day).

### 3.1. Comparison between Hardware Platforms

From the extensive tests performed in the literature [30] (see Table 1), we know that IBM’s *ibm_lagos* has a better performance than IonQ’s *Harmony* as far as mean fidelities for one- and two-qubit gates are considered. On the other hand, IonQ’s *Harmony* is preferable when state preparation and measurement (SPAM) fidelities are considered. More importantly, IonQ’s *Harmony* has the advantage of an all-to-all connectivity. This latter point is very relevant, because IBM’s quantum processors need SWAP gates to implement *D* (or SU(4)) gates between qubits that aren’t connected. Moreover, a SWAP gate is not a native gate on the IBMQ devices, and must be decomposed into three CNOT gates. Being the product of three CNOT gates, SWAP gates are expensive operations to perform on a noisy quantum device.

The results obtained using the direct method are shown in Figure 7, both for IBMQ and IonQ. As can be seen from the figure, particularly in the IonQ case, the purity of the whole state is greater than the bipartitions’ reduced purities, with the exception of the $n_{a}=1$ case for the IBMQ. This is equivalent to saying that the entropies of the parts are greater than the entropy of the whole state, which is a signature of bipartite entanglement in the system. Despite the fact that superconductor devices have lower error rates than IonQ for single-qubit and two-qubit gates, the overall purity is higher in trapped ion devices. This is mainly due to the complete connectivity of these machines, which allows avoiding noisy SWAP gates, in addition to the better SPAM fidelities of the ion-based device.

### 3.2. Entanglement Evolution

To investigate the survival of entanglement in an operating quantum computer, we iterate the above circuit for the generation of pseudo-random quantum states for a number of steps. In Figure 8, we consider *ibm_lagos* and n=4 qubits, and show the purities as a function of the number of steps, for subgroups of $n_{a}$ qubits. We can see that the purity of the overall system (ideally pure) is clearly higher than the purities of subsystems with $n_{a}=2$ and $n_{a}=3$ qubits up to 4 steps. For longer evolution times, the purity of the overall system drops below those of subsystems, and there is evidence, at least for $n_{a}=1,2$, of convergence to the purity for a maximally mixed state, equal to $1/2^{n_{a}}$. These values are smaller than those for pseudo-random states reported in Equation (Equation 5). Overall, the above remarks point to a vanishing entanglement content in the quantum hardware after 4–5 steps.

## 4. Conclusions

We investigated the generation of random states for which the entanglement content is almost maximal on a quantum computer. We proposed a method in which the obtained pseudo-random states converge to true random states by concatenating layers in which random permutations of the qubit labels are followed by the generation of random states for pairs of qubits. We argue that our method is optimal, and that the number of CNOT gates is greatly reduced with respect to circuits implementing two-qubit random unitary gates. The effectiveness of our method has been tested in the current implementations of quantum hardware, both for superconducting and ion trap quantum processors. In the latest implementation, we highlighted the advantages of the all-to-all connectivity of qubits.

With regard to the attainment of the maximal entanglement of quantum states, it would be interesting to study the class of maximally multipartite *n*-qubit states proposed by Facchi et al. [35]. More generally, multipartite entanglement optimization is a difficult task, which could at the same time be an ideal testbed for investigating the complexity of quantum correlations in many-body systems and for developing variational hybrid quantum-classical algorithms [36,37,38].

## Figures and Tables

**Figure 1 entropy-25-00607-f001:**
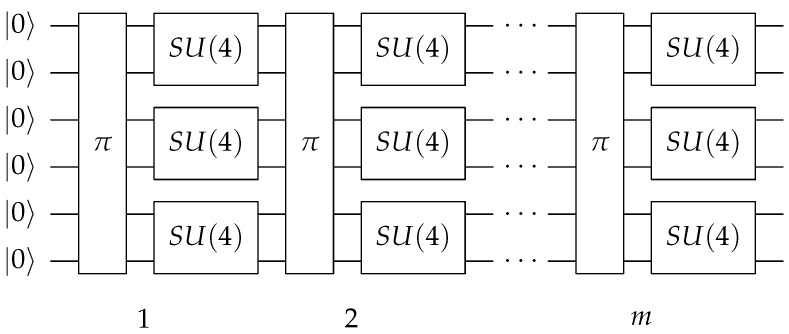
The pseudo-random state generator circuit consists of *m* layers of random permutations of the qubit labels, followed by random two-qubit gates. When the circuit width *n* is odd, one of the qubits is idle in each layer. In this figure, a circuit with n=6 qubits width is shown for illustration purposes.

**Figure 2 entropy-25-00607-f002:**
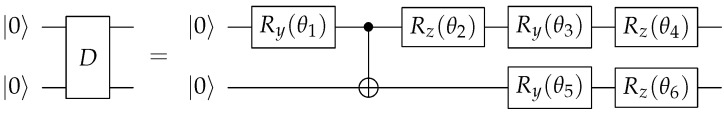
A circuit for two-qubit random state generation. Rotations $R_{k}$ are obtained by exponentiating the corresponding Pauli matrices $\sigma_{k}$.

**Figure 3 entropy-25-00607-f003:**
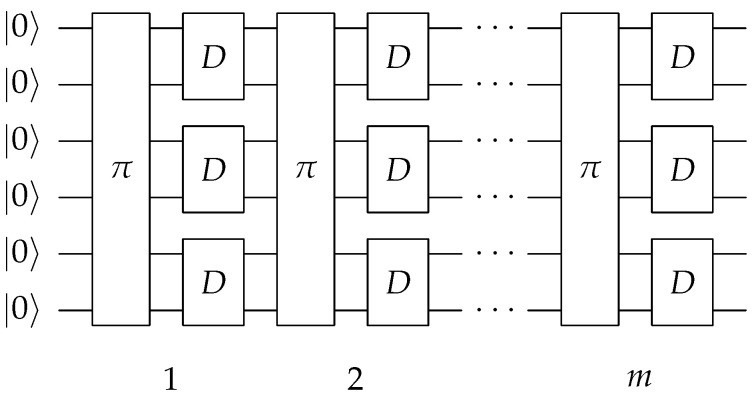
The pseudo-random state generator circuit consists of *m* layers of random permutations of the qubit labels, followed by random *D* gates. In this figure, a circuit with n=6 qubit’s width is shown for illustrative purposes.

**Figure 4 entropy-25-00607-f004:**
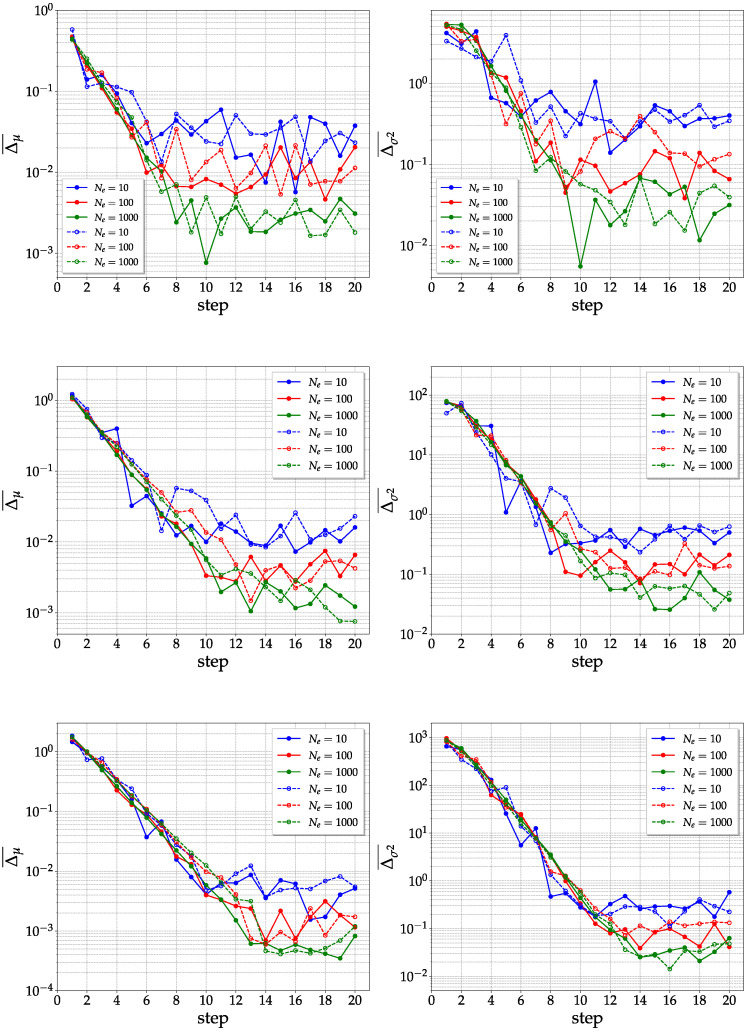
Average mean value relative error (**left**) and average variance relative error (**right**) for purities as a function of the number of steps (i.e., layers in the quantum circuit) and the ensemble size for 4-qubit (**top**), 6-qubit (**middle**), and 8-qubit (**bottom**) pseudo-random quantum state. The solid lines represent the direct method while the dashed lines represent the KAK method.

**Figure 5 entropy-25-00607-f005:**
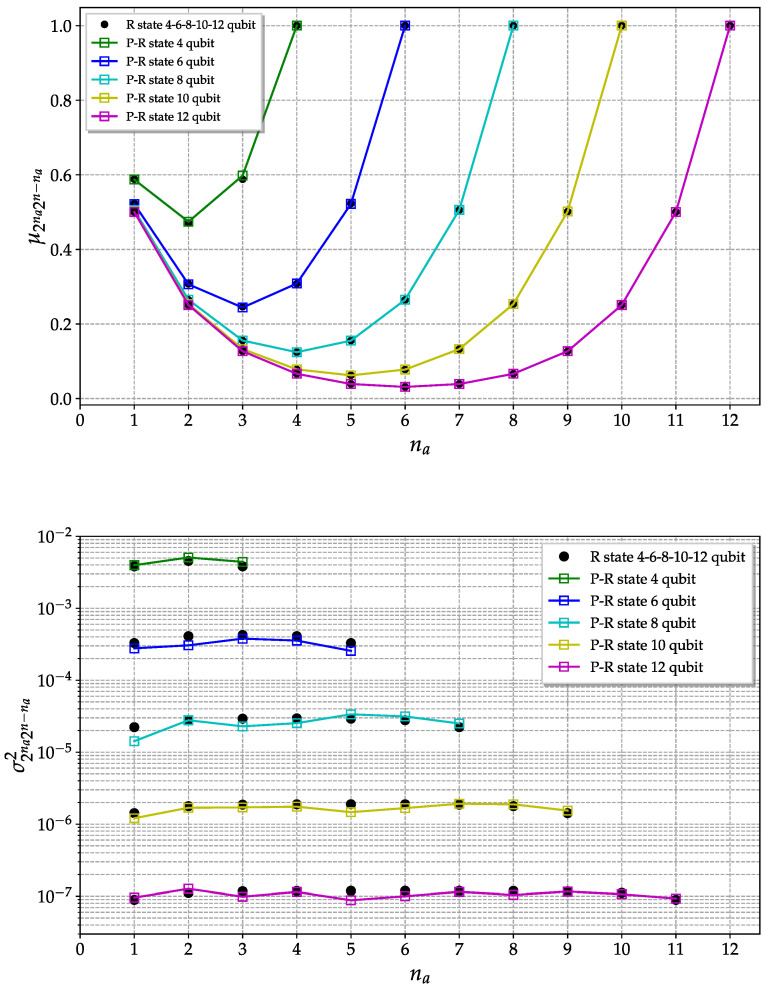
Purity mean value (**top**) and variance (**bottom**) of a pseudo-random quantum state plotted as a function of partition size. The various colors represent systems of different dimensions (number of qubits). The black dots are the expected values for a true random state. Here is shown the direct method with 20 steps and $N_{e}=100$.

**Figure 6 entropy-25-00607-f006:**
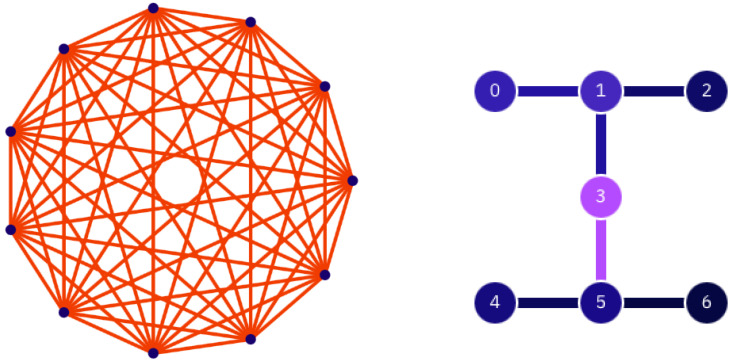
Architectures of the quantum processors used in this work. The circles represent the qubits while the lines represent the physical connection between them. On the left, we have the architecture of IonQ’s Harmony, which clearly shows the complete connectivity of ion-based devices. On the right, we have ibm_lagos. Here, the color scheme (blue for min, violet for max) refers to the single-qubit (color of the circles) and two-qubit (color of the lines) error rates. These are purely indicative since the rates change upon every calibration of the device.

**Figure 7 entropy-25-00607-f007:**
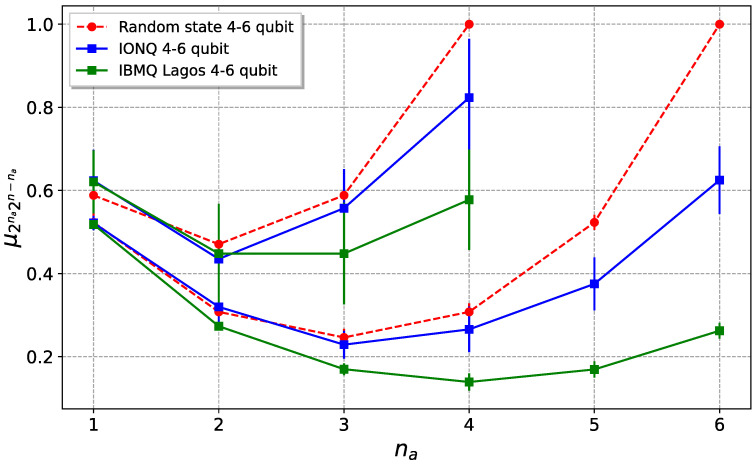
Comparison between the purities of a 4- and 6-qubit pseudo-random quantum states, generated in the two different realizations of a quantum computer investigated, with the direct method. In green, the superconductor IBM’s *ibm_lagos* is shown, while IonQ’s *Harmony* is shown in blue. Red curves give the results for ideal random states. Data were obtained on 10 September 2022, for *ibm_lagos* and on 24 July 2022, for *Harmony*.

**Figure 8 entropy-25-00607-f008:**
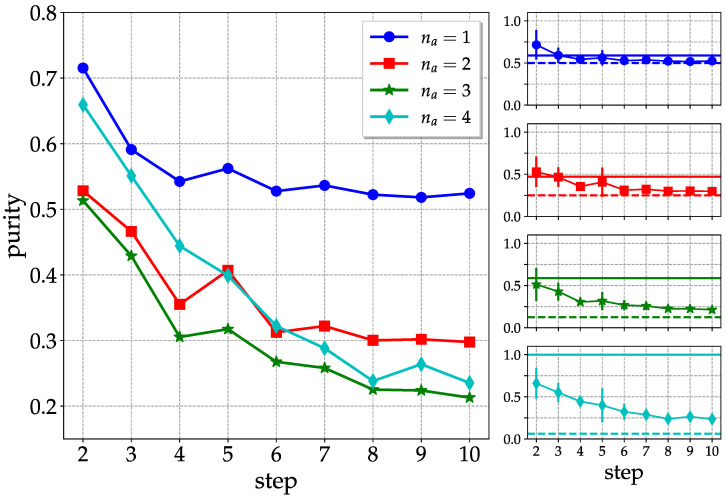
Evolution of the entanglement content of a pseudo-random quantum state generated by the circuit described in Figure 3 as a function of the number of layers (steps). The panels on the right show the individual curves, with the horizontal solid lines highlighting the purity expectation values for a true random state. The horizontal dashed lines refer to the purity of a maximally mixed state. Data taken from ibm_lagos on 29 January 2023.

**Table 1 entropy-25-00607-t001:** Table of quantum processing units (QPUs) evaluated in [30] using the quantum volume (QV) protocol. Values of QV, as well as single-qubit (1Q) gate, two-qubit (2Q) gate and state preparation and measurement (SPAM) fidelities are all vendor-provided metrics. The mean gate and SPAM fidelities are computed in [30] across all operations of the same type available on the device during the whole QV circuit execution duration. The number of edges for each backend was simply counted as the number of connections between qubits.

				QPU			Fidelity	
Vendor	Backend	QV	# Qubit	Topology	# Edges	2Q Gate	1Q Gate	SPAM
IBM Q	ibm_lagos	32	7	Falcon r5.11H	6	0.9924	0.9998	0.9862
IonQ	Harmony	8 *	11	All-to-All	55	0.96541	0.9972	0.99709

* The QV value for IonQ’s *Harmony* is the one measured in [30], since IonQ does not provide it.

## Data Availability

The dataset used and analyzed in the current study is available from the corresponding author upon reasonable request.

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
