# Peer review of "Generation of Pseudo-Random Quantum States on Actual Quantum Processors"

_entropy, 2023, doi:10.3390/e25040607_

Round 1

Reviewer 1 Report

I have carefully reviewed the paper describing a simplified method for generating multi-qubit random entangled states. The standard method involves using multi-layer circuits with SU(4) and random permutations, where SU(4) requires the synthesis of three CNOT gates. However, the proposed approach uses a direct method that only requires one CNOT gate.

The authors have evaluated the performance differences between the new and standard methods by measuring the average purity and variance of subsystems, and have demonstrated the results on both superconducting and ion-trap processors. The findings indicate that the performance of both methods is similar, and the ion-trap processor shows better overall performance due to its higher connectivity for random permutation gates.

The paper is well-written and clear. Therefore, I recommend its publication.

Author Response

We thank the reviewer for their interest in our work and positive assessment of our results.  

Reviewer 2 Report

The authors propose an efficient algorithm for the generation of pseudo-random states that makes use of random permutations and two-qubit gates. They first perform a numerical simulation of their circuit using Qiskit and show that its performance is essentially equivalent to the performance of a circuit using Cartan's KAK decomposition and needing a large number of CNOTs, which are usually the main source of errors in NISQ devices. Then they run the algorithm on IBM and IonQ quantum computers and compare their performances.

The paper is well written and  well structured. The technical aspects of the work appear valid and the results are of broad interest. Therefore, I recommend the article for publication in Entropy.

I have some minor comments and text editing suggestions.

Line 14: cloased --> closed

Line 49: distributions --> distribution

First line after equation (1):

are a suitable basis sets --> are suitable orthonormal sets

Line 101: random single qubits gates --> random permutations

Line 130:  [21] --> [20]

Line 140: measure of the state --> measure on the state space

Line 142: Reference missing

Line 144: Thus is --> This is

Lines 175, 190: tangle --> entangle 

Author Response

We thank the reviewer for their interest in our work and positive evaluation of our results.  
We are also grateful for the minor comments and editing suggestions, which we have implemented.

Reviewer 3 Report

The authors studied efficient methods of generating pseudo-random quantum states (of maximal entanglement), an arguably important and timely topic. The presentation is very clear and the theory well-explained with one crucial exception. In line 108 of page 4, the authors stated "...the problem now turns to find an efficient way (in sense that will be clarified later) to generate random SU(4) operators..." (bold mine). However, section 2.2 studies the direct method of generating two-qubit quantum states. This is a subtle change in the objective, and the efficiency gained is clearly explained by the authors in a parameter counting argument. Everything up to Figure 2 is absolutely correct, but from here onwards it is unclear to me how the two-qubit case can be  generalized to the n-qubit case. More specifically, the states after the first layer of D-gates in Figure 3 can be assumed to be a product of Haar-random two-qubit states, which contradicts the hypothesis of section 2.2, namely the initial starting state must be |00> before the second layer of D-gates can be applied. I presume this is the argument the authors have in mind; but I can be completely wrong here and another argument will work to extend the results to n-qubits. To reiterate, the authors made too big of a gap from (two-qubit) operators to states that seems to warrant a careful analysis. Of course the overall result is still correct and explain why the simulation works: the circuit in figure 3 will still converge to generating a random n-qubit state due to references [23-27] but with decreased efficiency. Therefore, I would like the authors to clarify this point before accepting the paper for publication.

Author Response

We thank the reviewer for finding "the presentation very clear and the theory well-explained".
We also appreciate their question. The key expectation by the reviewer is that "the circuit in figure 3 will still converge to generating a random n-qubit state due to references [23-27] but with decreased efficiency." With regard to convergence, we agree with the reviewer.
Indeed, in higher dimensions the entanglement content is highly 
typical, i.e. it is possible to show that the entanglement distribution for a random state becomes strongly peaked in the limit of a large number of qubits. This concentration of the measure explains the convergence of the method, improving for higher dimensions (see discussion at the end of page 7). The main point under discussion here is the efficiency of the Direct method. 
For instance, in Ref. [29] the use of CNOT gates turned out to be less efficient than the use of random two-qubit unitaries. In our case, the Direct method was found to be as efficient as the more expensive KAK decomposition of random SU(4) operations. Admittedly, this is just a numerical observation, lacking rigorous proof, and we have outlined this fact at page 2 and 6 (revised text in blue). 

Reviewer 4 Report

In this manuscript, the authors propose a method to generate pseudo-random quantum states on actual quantum processors. Near-term quantum computing devices (such as superconducting quantum processor [Science China Information Sciences 63, 180501 (2020)]) have the capability to enable some applications, as mentioned in the recent review paper [arXiv:2211.08737 (2022)]. Entanglement state is an important quantum resource for quantum computing to achieve quantum advantage. I could recommend the publication of this manuscript in Entropy after the authors address the following comment.

The proposed protocol seems the same as the protocol of quantum volume. Can the author clarify the difference between their work and quantum volume?

Author Response

We thank the reviewer for their interest in our work and for the question. Actually the quantum circuit we implement if analogous to the one used in quantum volume protocols. And indeed at page 2,  third paragraph, we mention the use of random quantum states for benchmarking quantum hardware. On the other hand, quantum circuits for validating quantum hardware typically use a heavy output generation problem for the observable bit string obtained at the end of the procedure, as described in Ref. [9] of our paper. In contrast, we provide a clear procedure to estimate the degree of multipartite entanglement in the quantum hardware. We believe this is a very important information, since, as the reviewer states, "Entanglement state is an important quantum resource for quantum computing to achieve quantum advantage". Furthermore, in our paper we propose a method to efficiently generate pseudo-random quantum states and use it to benchmark actual quantum hardware, both for superconducting and ion trap quantum processors.

Round 2

Reviewer 3 Report

I appreciate the authors' response. It would be great if "numerical evidence" can be included somewhere in the abstract to better reflect the results of the paper. I leave this completely at the discretion of the authors.